# Qualitative Code Suggestion:
# A Human-Centric Approach To Qualitative Coding

**Cesare Spinoso-Di Piano[1,2]**
**Samira Abbasgholizadeh Rahimi[1,2]***    **Jackie Chi Kit Cheung[1,2]**
McGill University[1]    Mila Quebec AI Institute[2]
{cesare.spinoso-dipiano@mail., samira.rahimi@, jackie.cheung@}mcgill.ca

## Abstract

Qualitative coding is a content analysis method in which researchers read through a text corpus and assign descriptive labels or *qualitative codes* to passages. It is an arduous and manual process which human-computer interaction studies have shown could greatly benefit from NLP techniques to *assist* qualitative coders. Yet, previous attempts at leveraging language technologies have set up qualitative coding as a fully automatable classification problem. In this work, we take a more assistive approach by defining the task of *qualitative code suggestion* (QCS) in which a ranked list of previously assigned qualitative codes is suggested from an identified passage. In addition to being user-motivated, QCS integrates previously ignored properties of qualitative coding such as the sequence in which passages are annotated, the importance of rare codes and the differences in annotation styles between coders. We investigate the QCS task by releasing the first publicly available qualitative coding dataset, `CVDQuoding`, consisting of interviews conducted with women at risk of cardiovascular disease. In addition, we conduct a human evaluation which shows that our systems consistently make relevant code suggestions.

## 1 Introduction

In qualitative research, qualitative coding is a content analysis method that is used to analyze textual corpora such as cultural and political texts, questionnaires and interview transcripts. First used in social science studies, qualitative coding has become a ubiquitous and universal method of analysis geared at providing researchers with an in-depth understanding of their studied corpus (Elliott, 2018). It is now employed in several research disciplines including medicine, education, and psychology (Chapman et al., 2015; Gough and Scott, 2000; Smith and McGannon, 2018). During the

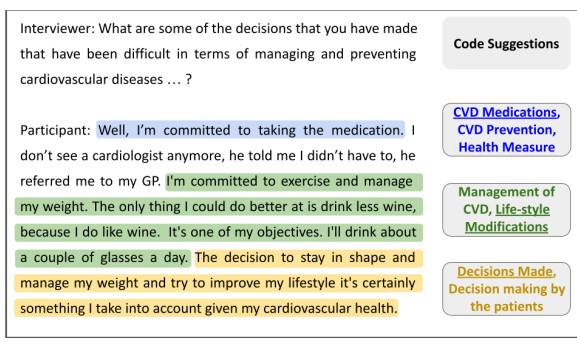

Figure 1: Excerpt from one of our interview transcripts along with qualitative code suggestions made by one of our systems (GPT-3.5). The codes that were originally assigned to the passages by the qualitative coder are underlined.

coding process, a researcher carefully scans each document in their corpus, identifies passages that are associated with their research question, and assigns these passages descriptive labels or *qualitative codes* (Figure 1).

Despite its widespread use, coding remains an arduous and time-consuming process as it requires manually annotating a studied corpus line by line. For instance, studies have shown that coding a 1-hour interview transcript will typically take an experienced researcher between 4 to 8 hours of annotation (Miles et al., 2019).

To increase coding efficiency and to reduce its cognitive load on researchers, NLP practitioners have attempted to insert language technologies into the coding process by formulating it as a text classification task. However, as human-computer interaction (HCI) studies have shown (Rietz and Maedche, 2021), qualitative coders prefer receiving suggestions for identified passages as they code rather than having yet-to-be-seen passages automatically coded for them. This aversion towards a fully automatic coding system may explain why, to this day, all available qualitative coding tools are used solely as "electronic filing cabinets" (Fielding and

---

*Corresponding author

Lee, 2002). That is, if automatic coding is not approached in a user-centered manner, despite a desire for automated assistance, coding tools such as NVivo[1] and MAXQDA[2] will continue to be used for their bookkeeping features only (Marathe and Toyama, 2018).

In this work, we take a more user-centered approach to qualitative coding by proposing the task of *qualitative code suggestion* (QCS) in which a ranked list of previously assigned codes is suggested from passages identified by a qualitative coder. Our framing of qualitative coding reveals technical subtleties which have not been addressed in previous work. For instance, our task definition exposes the necessity to evaluate code suggestions based on the sequence in which documents are coded as well as the importance, and difficulty, of detecting rare codes. In addition, unlike other NLP tasks that use human-sourced annotations, QCS expects differences in annotation styles and avoids attempts to correct or homogenize these differences. Although more challenging, we believe that this *human-centric* approach is appropriate given that qualitative coding is meant to aid a researcher in their *personal* understanding of a corpus.

To summarize, the research questions we address in our work are: 1) What previously unexplored technical challenges arise from approaching qualitative coding in a user-centered manner? and 2) How effective are current NLP methods at handling these challenges? To this end, the contributions of our work are as follows:

- Grounded in HCI studies on automated coding assistance, we propose the task of qualitative code suggestion (QCS) in which previously assigned codes are suggested from passages identified by qualitative researchers. This user-centered approach to qualitative coding reveals the importance of considering the sequence in which passages are coded, detecting rare codes, and preserving differences in coders' annotation styles.

- We release the first publicly available qualitative coding dataset, named CVDQuoding, consisting of transcripts of interviews with women at risk of cardiovascular diseases along with the annotations of two qualitative

coders. This dataset is available by request to the corresponding author, due to its sensitive nature.

- We experiment with classification, information-retrieval and zero-shot prompting techniques to model QCS and conduct a human evaluation which shows that our systems consistently make relevant code suggestions.[3]

## 2 Related Work

In qualitative research, content analysis is defined as "a family of research techniques for making systematic, credible, or valid and replicable inferences from texts and other forms of communications" (Drisko and Maschi, 2016). These techniques rely heavily on qualitative coding which has been used extensively throughout different areas of research, first in the social sciences and, now, across most fields (Macnamara, 2006). There are two types of qualitative coding: deductive and inductive (Saldaña, 2021). In deductive coding, all codes are predefined and the coding process serves as a way of validating or invalidating a hypothesis. In inductive coding, codes are defined ad hoc as documents are analyzed and, thus, the coding process is used to discover new phenomena underlying a corpus.

In terms of assistive technologies for qualitative coding, studies have investigated various computational approaches to automate parts of the *deductive* qualitative coding process. Efforts by Crowston et al. (2012); Scharkow (2013); McCracken et al. (2014); Liew et al. (2014) have investigated the use of rule-based learning and supervised learning to automate the assignment of codes to passages. More recently, Grandeit et al. (2020); Xiao et al. (2023) have explored using BERT-based models and GPT3 respectively to predict a previously assigned code from a passage. In all cases, deductive coding was cast as a fully automatable classification problem and metrics such as F1 and annotator-system agreement scores were used to measure performance.

At the same time, there has been extensive work from the HCI community on how qualitative coders would like automation to assist them during the coding process (Marathe and Toyama, 2018; Feuston and Brubaker, 2021; Chen et al., 2018; Haug et al.,

---

[1] https://lumivero.com/products/nvivo/
[2] https://www.maxqda.com/qualitative-data-analysis-software

[3] The code can be found at https://github.com/cesare-spinoso/QCS.git

2021). These studies show that while researchers desire automation, they believe the coding agency should remain in their hands. Thus, methods that introduce language technologies in the qualitative coding process should favour assisting coders by highlighting ambiguous passages (Drouhard et al., 2017), by helping them remain consistent in their annotation style (Sporer, 2012), or by interactively refining the definition of codes (Bakharia et al., 2016). These assistive approaches differ from previous human-in-the-loop annotation studies (Montani and Honnibal, 2018) since, as Jiang et al. (2021a) point out, "qualitative analysis is a unique case for human-AI collaboration because uncertainty is essential to qualitative analysis itself." In other words, qualitative coding is unique because the act of coding data, and the uncertainty that comes with it, is valued by researchers as it cultivates their understanding of their corpus.

## 3 Qualitative Code Suggestion

We present properties of qualitative coding which have surfaced from the user-centered perspective we take in this work. We show that these properties warrant a new task definition that centers around suggestions and rare codes.

### 3.1 Properties of Qualitative Coding

In qualitative coding, researchers code their corpus to gain a deeper understanding of a certain phenomenon related to their research question. As a result, in deductive and inductive coding, the first few documents that are coded are critical for the researcher's *own* understanding. Once the researchers have read through enough documents, properties of qualitative coding arise which, according to HCI studies (Rietz and Maedche, 2021; Marathe and Toyama, 2018), should be considered when developing assistive coding techniques. We further develop these properties below:

1. *Data saturation.* Coders that carry out qualitative coding reach a personal point of *data saturation* after which few new codes will appear (Saunders et al., 2018). It is at this point where suggesting relevant codes for an identified passage may be helpful for a qualitative coder. These suggestions could help the qualitative coder consider relevant code assignments without needing to sift through irrelevant codes. At the same time, suggesting several relevant codes minimizes the shift

towards complete automation as coders could still reject suggestions.

2. *Rare codes.* Even though new codes may be rare past the point of data saturation, their occurrences remain possible and scientifically important to the coders (van Rijnsoever, 2017). That is, the rarity of a code is in and of itself an important scientific finding. Thus, while suggesting relevant codes may alleviate some of coding's cognitive load, detecting *rare codes* may be just as important.

3. *Annotation style.* Finally, since coding is meant to cultivate a particular researcher's understanding of a corpus, no two coders will have the same *annotation style* (Loslever et al., 2019). Thus, previous efforts to homogenize code annotations, for instance by grouping similar codes from different coders together, should be avoided. Although convenient for modeling purposes, this merging is not supported by the user-centered approach we take in this work.

### 3.2 Task Definition

Our task definition, which is applicable to both deductive and inductive coding, integrates the previously discussed properties into the main QCS task and the novel code detection subtask.

**Qualitative code suggestion (QCS)** Consider the setting in which a corpus is comprised of $N$ documents $\mathcal{D} = \{d_1, d_2, \ldots, d_N\}$. Each document, $d_i$, is a set of text spans $\{t_n\}_{n=1}^{\text{len}(d_i)}$ which have been identified and assigned codes by a qualitative coder. Now, suppose $K \in \{1 \ldots N\}$ is a *particular* coder's point of *data saturation*. In this case, we collect all the codes assigned to text spans from documents $d_1$ to $d_K$ and create a set of codes $\mathcal{C} = \{c_1, \ldots, c_m\}$. The task of QCS that we define here consists of ranking, in order of relevance, the set of codes in $\mathcal{C}$ with respect to every identified text span in the set of documents past the point of data saturation, i.e. $\{t \in d_i : i = K + 1 \ldots N\}$.[4]

**Novel code detection subtask** Furthermore, since documents are coded sequentially, some rare codes may only surface in the test set portion of the corpus, $\{d_{K+1}, \ldots, d_N\}$, and, thus, might not

---

[4]We leave the additional task of text span extraction as future work as this extension to QCS goes beyond the scope of this study.

be part of $\mathcal{C}$. As detecting rare codes may be just as important as suggesting relevant ones, we add a *novel* code detection subtask to our task definition which consists of assigning the novel code to a text span if one of its truly assigned codes is not in $\mathcal{C}$. To create training instances for this subtask, we assign the novel code to all text spans in the training set that have been assigned a code with a training frequency of 1. All of the text spans in the validation and test sets with at least one assigned code that does not appear in the training set are assigned the novel code.

## 4 The `CVDQuoding` Dataset

To investigate the QCS task, we release the first publicly available `qualitative coding` dataset which we name `CVDQuoding`. In this section, we describe the dataset and how it was created. In addition, we draw parallels between `CVDQuod-ing` and the properties of qualitative coding we described in Section 3.1.

`CVDQuoding` is a dataset consisting of 15 transcripts of interviews conducted with women at risk of cardiovascular diseases (CVDs) as well as annotations carried out by two qualitative coders. These qualitative coding annotations consist of text spans identified by the qualitative coders, as well as the codes that were assigned to each text span. Additional details about the dataset as well as comparisons with previous qualitative coding datasets, which have all been closed-source, can be found in Table 4 in Appendix A.1.

The annotations in `CVDQuoding` were originally created as part of a larger project to investigate the interest of women at risk of CVDs in using AI tools (Bousbiat et al., 2022) and were subsequently adapted for our purposes. During the interviews, women were asked about the health challenges they face and their opinions about using an AI tool to help them manage their risk of CVDs. For this work, we extracted the 15 interview transcripts and tagged the text spans coded by each coder and the questions asked by the interviewer using XML. Readers may refer to Table 6 in Appendix A.2 for the complete list of questions asked during each interview and to Figure 6 in Appendix A.1 for an example of a tagged transcript excerpt.

`CVDQuoding` exhibits the properties of qualitative coding presented in Section 3.1. In terms of *data saturation*, both coders create many new

codes in the first five interviews, and this trend tapers off as they code more transcripts (Figure 2). In addition, coder 1 finds new *rare codes* in every transcript whereas coder 2 ceases to create new codes by interview 7. Moreover, both coders clearly exhibit significantly different *annotation styles* with coder 1 creating 207 codes with an average of 4 text spans per code and coder 2 creating 23 codes with an average of 19 text spans per code. This difference can be attributed to the specificity of coder-2's codes, which tend to be more abstract (e.g., "Health Measures Taken") than coder-1's codes (e.g., "CVD-related examinations, tests or measures"). For additional details about `CVDQuoding`, including a sample list of codes, see Appendix A.1.

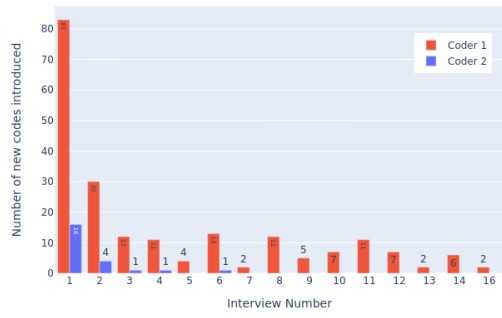

Figure 2: Distribution of the number of new codes introduced per interview for each coder.

## 5 Modeling Approaches

We investigate the ability of current NLP methods to rank lists of codes by relevance as well as detect novel codes from text spans. To this end, we describe our data formatting decisions and the modeling paradigms we explore.

We choose to explore three distinct modeling paradigms, classification, information-retrieval and zero-shot prompting, based on the following intuitions. Firstly, the classification paradigm assumes that the confidence scores of a multi-label classifier can be used to rank codes. Secondly, the information-retrieval paradigm assumes that codes can be treated as documents and that text spans are the queries used to rank them. Finally, the zero-shot prompting paradigm assumes that the descriptiveness of codes is sufficient for a large language model (LLM) to rank them based on the text span.

## 5.1 Data Formatting

For all three modeling paradigms, we associate every contiguous text span $t \in d_i$ that was coded by one of the qualitative coders with its immediate context $c$. In the case of `CVDQuoding`, we define the context $c$ as being the question asked right before $t$. Thus, in our case, every model has access to a tuple $(q, t)$ consisting of a question, $q$, and a text span, $t$, highlighted by a qualitative coder. We use the previous question as an approximation of the context as most coding decisions in `CVDQuoding` involve considering a text span as the answer to a preset question. For example, the code "Interest in AI tools" is assigned to "Very interested, yes" because it is the answer to "Are you interested in using AI tools for CVD?".

## 5.2 Classification Paradigm

We build a $|\mathcal{C}| + 1$ multi-label binary classifier trained on the first $K$ annotated transcripts to predict the assignment of each code $c \in \mathcal{C}$ as well as the novel code to a text span. We use the scores produced for each code to sort the set of codes $\mathcal{C}$ for a test instance $(q, t) \in \{(q, t) : t \in d_i, i = K+1 \ldots N\}$. Furthermore, an instance is assigned the novel code only if it is given the highest score among all codes.

## 5.3 Information-retrieval Paradigm

In the information-retrieval paradigm, we treat an instance $(q, t)$ as a query and use two neural-retrieval architectures to rank the set of codes $\mathcal{C}$ with an additional step for the novel code detection subtask.

We build two neural-retrieval architectures originally presented by Reimers and Gurevych (2019): the bi-encoder and the cross-encoder. In the bi-encoder, a representation is learned for the text span $t$, the previous question $q$ and the code $c$. The representations of the test span, $h_t$, and the previous question, $h_q$, are max pooled together and a score is computed by applying a cosine similarity between the pooled representation and the code representation, $h_c$. In the cross-encoder, representations are learned for the concatenations of the code with the question, $q$ [SEP] $c$, and with the text span, $t$ [SEP] $c$. The representations, $h_{q\,[\text{SEP}]\,c}$ and $h_{t\,[\text{SEP}]\,c}$ are max pooled together and a classification head is placed on top of the pooled representation to produce scores. In both cases, the code scores produced are used to rank all but the novel code.

For the novel code detection subtask, a classification head is trained on top of the vector of scores $\hat{\mathbf{y}}_\mathbf{i}$ which consists of the scores computed between each code $c \in \mathcal{C}$ and the input instance $(q, t)_i$. Thus, if there are 10 codes in the training set (excluding novel), then for every instance $(q, t)_i$ the classifier is passed a 10-dimensional vector $\hat{\mathbf{y}}_\mathbf{i}$ computed from either the bi-encoder or cross-encoder.

## 5.4 Zero-shot Prompting Paradigm

In the zero-shot prompting paradigm, we provide an autoregressive LLM $M$ with a prompt containing general instructions, the list of codes $\mathcal{C}$ and a text span $t$ to code along with its previously asked question $q$. Refer to Appendix A.3 for the prompt template. Upon generation, the suggested codes are extracted via an exact-match search and the order in which the codes are generated is used as their predicted rank. Furthermore, the generation "predicts" the novel code either if $M$ generates the string "None of the above" first (which is included in the prompt template) or if no exact matches are found in its generation.

## 6 Experiments

We investigate our approaches to modeling the QCS task in order to determine the ability of current NLP-based methods to allow for a more user-centered approach to qualitative coding. We discuss the experimental setup to test our modeling paradigms as well as our evaluation methodology.

### 6.1 Experimental Setup

To experiment with each of the paradigms presented in the previous section, we sort `CVDQuoding` by annotation-time, group it by coder, and consider it at different possible points of data saturation. More specifically, for each coder, we consider training on $\{d_1, \ldots, d_K\}$ and testing on $\{d_{K+1}, \ldots, d_{15}\}$ for $K = 1 \ldots 14$ where $d_i$ is the $i^{\text{th}}$ annotated transcript in annotation-time. We reserve 20% of each training set for validation. We summarize our workflow in Figure 3.

We train and test models from each of our three modeling paradigms and for all of the configurations discussed above. For the classification paradigm, we use $|\mathcal{C}| + 1$ SVM classifiers as well as a single DistilBERT (Sanh et al., 2020) classifier with a multi-label sequence classification head. We use DistilBERT due to the computational costs

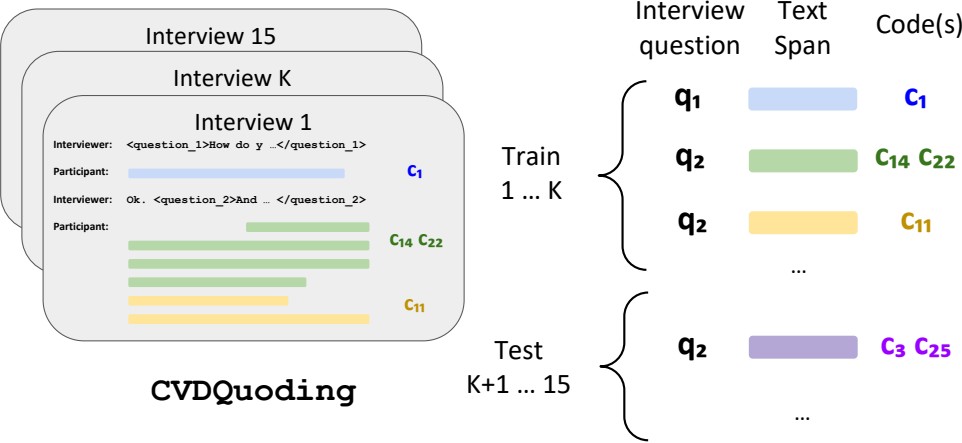

Figure 3: Visualization of the data formatting we apply to the raw annotations from CVDQuoding to run our experiments with our different modeling paradigms.

of hyperparameter tuning over $2 \times 14 = 28$ individual datasets. For the information-retrieval paradigm, we use DistilBERT as the encoder for the bi-encoder architecture. For the cross-encoder, we experiment with both DistilBert and ConvBERT (Jiang et al., 2021b). We use ConvBERT based on the intuition that text spans often contain phrases which are lexically similar to a code's description. If this is the case, then ConvBERT's span-based kernels may be better suited at soft matching a code's description in a text span than a fully attention-based masked language model like DistilBERT. Moreover, ConvBERT has computational costs (e.g., GPU memory requirements) in the same order of magnitude as DistilBERT. Finally, for our zero-shot prompting paradigm, we use OpenAI's GPT-3.5 Turbo (Brown et al., 2020) accessible through its API[5]. Additional hyperparameter configurations and training details can be found in Appendix A.3.

In addition to the models discussed above, we use an information-retrieval baseline commonly used in neural-retriever papers. We use the Okapi BM25 retriever (Trotman et al., 2014) to compute scores between each code and instance $(q, t)$. In addition, we place a logistic regression model on top of the vector of scores to make the novel class prediction.

## 6.2 Evaluation Methodology

We subject our modeling paradigms to two distinct rounds of evaluations. In Section 6.2.1, we evaluate how well our systems provide code suggestions based on the original annotations from CVDQuoding. In Section 6.2.2, we further evaluate our systems suggestions by asking human judges to determine their relevance. We present a visualization of our two-phase evaluation in Figure 4.

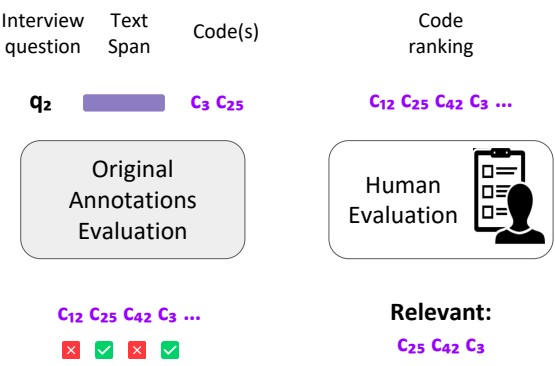

Figure 4: The original annotations evaluation relies on the codes assigned to the test text spans in CVDQuoding while the human evaluation asks judges to determine which of the top-4 suggested codes are relevant (for a sample of test instances).

### 6.2.1 Original Annotations Evaluation

**QCS** For the main QCS task, we compute the mean reciprocal rank (MRR) score and a soft normalized discounted cumulative gain at $k$ (sNDCG@k) score. These metrics allow us to

[5]https://platform.openai.com/docs/api-reference

measure how highly our systems rank the codes that were originally assigned to the text spans in `CVDQuoding` by the qualitative coders.

To compute the MRR, we exclude the rank of the novel code as we are interested in a system's ability to suggest previously assigned codes. Additional details about the computation of the MRR are presented in Appendix A.4.

To compute the sNDCG@k metric, we approximate a suggested code's $c$ relevance score for a text span $i$, $rel_i(c)$, and subsequently carry out the standard NDCG computation. The relevance score, $rel_i(c)$, is computed by calculating the BERTScore (Zhang et al., 2019) between suggested and true codes. More specifically, the relevance score $rel_i(c)$, for a code $c \in \mathcal{C}$ and a text span $i$ is

$$rel_i(c) = \max_{c' \in C_i^{\text{true}}} \text{BERTScore}(c, c')$$

where $C_i^{\text{true}}$ is the set of codes originally assigned to a text span $i$ by one of the two qualitative coders.

Once all the relevance scores are computed, we use them in the standard NDCG computation. To do so, we sort the list of relevance factors $(rel_i(c) : c \in \mathcal{C})$ to compute the `true_rank_scores` list for a text span $i$. In addition, the `predicted_rank_scores` for a text span $i$ is computed as the list of relevance factors sorted by the model's scores for each code. Both these lists are then used in the standard NDCG formula. Additional details are presented in Appendix A.4.

Finally, we add a cutoff $k$ to sNDCG as is usually done in standard NDCG to account only for the top $k$ results creating sNDCG@$k$. We choose to evaluate our systems with $k = 4$ as no more than 4 codes are assigned to text spans throughout `CVDQuoding`.

**Novel code detection** For the novel code detection subtask, we compute both the macro and micro $F_1$ scores. We report both averages due to the imbalance in the ratio between novel and not-novel classes.

### 6.2.2 Human Evaluation

In addition to using `CVDQuoding`'s original annotations to evaluate our systems' ability to provide relevant code suggestions, we conduct a human evaluation where we ask human judges to ascertain the relevance of our systems' code suggestions. This additional evaluation phase is necessary because it is common for text spans to have several alternative and relevant code assignments which the original coders may not have considered.

**Evaluation setup** To gather human judgements, we hired two human evaluators, both with experience in qualitative research, and asked them to judge the relevance of our systems' code suggestions. In particular, we randomly sample 32 instances from the annotations of coders 1 and 2 at the same point of data saturation of $K = 10$ and assign a human evaluator to each sample. We extract the top-4 suggestions of each model described in Section 6.1[6] and ask the evaluator to judge whether each suggestion is "Relevant" or "Irrelevant" based on the question and the text span. Additional details about the human evaluation can be found in Appendix A.5.

**Metrics** We use the annotations from the human evaluation to recompute the rank-based metrics presented in Section 6.2.1. We recompute the MRR and the sNDCG@k of the 32 code suggestions based on the human evaluator's relevance judgements. That is, for a sampled instance $i$, $C_{\text{true}}^i$ becomes the set of codes selected as "Relevant" by the evaluator. In addition, we also compute the precision at 4 (P@4) of each system using the human evaluator's annotations. Details about the computation of P@4 are presented in Appendix A.4.

## 7 Results

We present the results of our experiments with respect to both the original annotations and the human evaluation annotations.

### 7.1 Original Annotations

Using the original annotations as the gold standard labels, we present the MRR and sNDCG@4 scores for the main ranking task of QCS as well as the macro and micro $F_1$ for the novel code detection subtask. In particular, we show the MRR and sNDCG@4 scores at $K = 10$ for coders 1 and 2 (Table 1) and the $F_1$ scores for the novel code prediction task at $K = 5$ (Table 2). In addition, we provide plots of the MRR for coders with respect to $K$ (Figure 5 and in Appendix A.6). Additional results for all points of data saturation, presented using both scatter plots and tables, can be found in Appendix A.6.

---

[6]We do not consider the Okapi BM25 baseline suggestions due to their poor quality.

|  | MRR | | sNDCG@4 | |
| --- | --- | --- | --- | --- |
|  | Coder 1 | Coder 2 | Coder 1 | Coder 2 |
| Okapi BM25 | 0.15 | 0.53 | 0.70 | 0.79 |
| SVM | 0.51 | 0.75 | 0.80 | 0.85 |
| DistilBERT | 0.55 | 0.75 | 0.70 | 0.79 |
| Bi-Encoder | 0.48 | 0.74 | 0.80 | 0.86 |
| Cross-Encoder (DistilBERT) | 0.55 | 0.64 | 0.81 | 0.82 |
| Cross-Encoder (ConvBERT) | **0.59** | **0.77** | **0.83** | **0.86** |
| GPT-3.5 | 0.57 | 0.73 | 0.74 | 0.83 |

Table 1: Results of the models' performance on the QCS main ranking task. The MRR and sNDCG@4 scores are computed based on the original annotations from the `CVDQuoding` dataset and using $K = 10$ as the point of data saturation.

|  | Macro F1 | | Micro F1 | |
| --- | --- | --- | --- | --- |
|  | Coder 1 | Coder 2 | Coder 1 | Coder 2 |
| Okapi BM25 | **0.56** | **0.57** | **0.59** | **0.80** |
| SVM | 0.32 | 0.30 | 0.44 | 0.30 |
| DistilBERT | 0.55 | 0.45 | **0.59** | **0.80** |
| Bi-Encoder | 0.36 | 0.45 | 0.57 | **0.80** |
| Cross-Encoder (DistilBERT) | 0.51 | 0.46 | **0.59** | 0.78 |
| Cross-Encoder (ConvBERT) | 0.36 | 0.57 | 0.57 | 0.79 |
| GPT-3.5 | 0.41 | 0.44 | 0.53 | **0.80** |

Table 2: Novel class detection subtask results. Macro and micro $F_1$ scores are computed using $K = 5$.

**The information-retrieval and zero-shot prompting paradigms are the best performing** On the main QCS ranking task, we observe that the information-retrieval paradigm, modeled through the ConvBERT cross-encoder, and the zero-shot prompting paradigm, modeled through GPT-3.5, achieve the highest MRR and sNDCG@4 scores across both coders for most data saturation points. While GPT-3.5 achieves the highest MRR in most cases for coder 1 (11 out of 14), ConvBERT obtains the largest number of maximal MRR scores for coder 2 (7 out of 14). In addition, ConvBERT obtains the largest number of maximal sNDCG@4 scores for both coders.

**Detecting novel codes is a challenging subtask** For the novel code detection subtask, no modeling paradigm is able to outperform the Okapi BM25 baseline. These relatively poor performances indicate that detecting novel codes is a challenging subtask.

## 7.2 Human Evaluation

Using the human evaluation annotations as the gold standard labels, we present the results for the same rank-based metrics, along with the added P@4 results, and contrast them with the results in the previous section (Table 3).

**Systems consistently make several relevant code suggestions** While the human evaluators have a recall with respect to the original annotations of 0.94 and 0.95 for coders 1 and 2 respectively, they identify, on average, 3 times more relevant codes than in the original annotations. As a result, we observe a statistically significant jump between the MRR and sNDCG@4 scores computed on the sample of 32 text spans using the original annotations and the human evaluation annotations across all models. On average, the human-evaluation-derived MRR is 66% and 34% higher than the MRR computed using the original annotations for coders 1 and 2, respectively. In fact, in the case of coder 2, the recomputed MRR for GPT-3.5 is 1. This jump is less drastic for the sNDCG@4 metric with relative increases of 40% and 19% for coders 1 and 2 respectively. In addition to making consistently relevant code suggestions, our best performing models achieve P@4 scores above 0.5 indicating that, on average, our systems are able to suggest more than one relevant code for a given passage. Altogether, these additional results demonstrate that human evaluations are necessary for QCS systems as they provide interpretations for codes which were highly ranked but which the qualitative coders may not have originally considered in their code assignments.

|  | MRR | | sNDCG@4 | | P@4 | |
| --- | --- | --- | --- | --- | --- | --- |
|  | Coder 1 | Coder 2 | Coder 1 | Coder 2 | Coder 1 | Coder 2 |
| SVM | 0.80 | 0.93 | 0.970 | 0.978 | 0.40 | 0.55 |
| DistilBERT | 0.72 | 0.91 | 0.969 | 0.975 | 0.41 | 0.65 |
| Bi-Encoder | 0.86 | 0.90 | 0.972 | 0.979 | 0.50 | 0.66 |
| Cross-Encoder (DistilBERT) | 0.78 | 0.93 | 0.965 | 0.982 | 0.46 | **0.68** |
| Cross-Encoder (ConvBERT) | **0.87** | 0.95 | **0.973** | 0.987 | **0.51** | 0.56 |
| GPT-3.5 | 0.74 | **1.00** | **0.973** | **0.989** | 0.32 | 0.53 |

Table 3: MRR, sNDCG@4 and P@4 scores computed using the annotations collected during the human evaluation.

## 8 Analysis

We discuss the implications of our results in relation to the three properties of qualitative coding which we uncovered through our human-centric ap-

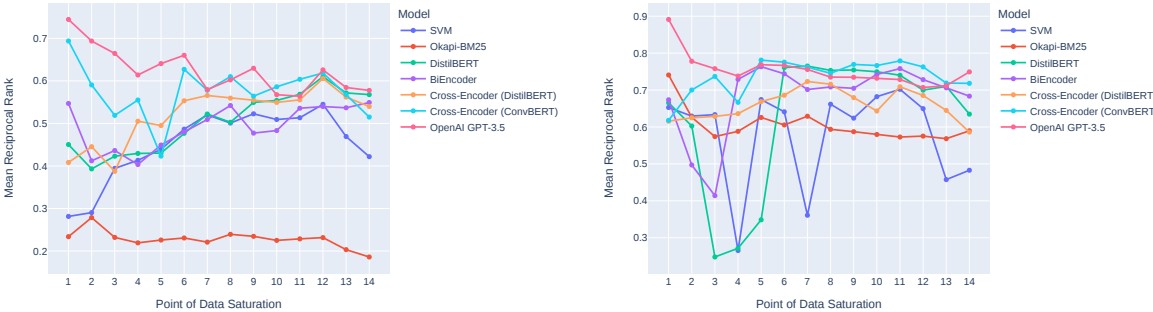

Figure 5: Plot of MRR across data saturation points $K = 1$ to 14 for coder 1 (left) and for coder 2 (right)

proach and the ability of our modeling paradigms to consider these properties.

Firstly, across both coders and all models, we observe a steady rise in our systems' ability to make code suggestions consistent with the original annotations of `CVDQuoding` (See scatter plots through annotation-time in Figure 5 as well as in Appendix A.6). This poor performance for small values of $K$ is consistent with inductive coding which, for the first documents, is exploratory and unpredictable by nature. In fact, the moment at which this rise in performance plateaus may be a helpful signal to identify when a coder has reached their personal point of *data saturation* and, thus, when code suggestions could start to be beneficial. Secondly, across all saturation points and for both coders, we observe relatively poor performance in detecting novel codes. This difficulty suggests that the "catch-all" bucket method we use in this work is inappropriate and that more sophisticated representations of novelty need to be learned to properly detect *rare codes*. Finally, we notice that, on average, our systems' MRR and the sNDCG@4 calculated using coder 2's annotations are 58% and 7% higher respectively than when calculated using coder 1's annotations. This result is natural given coder 2's *annotation style* of assigning high-level codes to larger text spans is much more amenable to machine learning techniques. Thus, in the future, methods that are able to identify annotation styles and incorporate them in their modeling may be more suitable at solving QCS in a user-tailored fashion.

In addition, the ranked-based metrics computed using human judgments point to a promising future for the downstream viability of QCS systems. The recomputed MRR and the additional results from the P@4 show that not only are our systems able to correctly rank relevant codes, but they are also likely to highly rank more than one relevant code, potentially forcing the coder to further reflect on their coding decisions without needing to consider irrelevant codes.

## 9  Conclusion

In this work, we approached qualitative coding in a novel user-centered manner and showed that this approach gives rise to technical subtleties related to qualitative coding which have not been previously investigated. These subtleties include the importance of considering a coder's personal point of data saturation, the difficulty of detecting rare codes and the necessity to avoid homogenizing different annotation styles. We showed that each of these properties is reflected in both the `CVDQuoding` dataset we introduced and the experimental results of our modeling of QCS. Lastly, our human evaluation showed that solely relying on automatic metrics computed with respect to the original annotations made by the qualitative coders is insufficient to describe the performance of QCS systems. The results derived from our human evaluation show that QCS systems can consistently provide several relevant code suggestions and, thus, that this human-centric approach to qualitative coding may be able to truly *assist* researchers in their study of textual corpora.

## Limitations

This work includes a few limitations which we leave as future work. Firstly, we did not conduct

an evaluation of the impact of QCS in an applied setting. This lack of downstream experiments prevents us from determining the full impact of automatically providing code suggestions to qualitative coders. For instance, an automatic QCS system may remove more agency from the coder than anticipated if they begin to blindly trust suggestions. Secondly, despite our best efforts, our work falls short of investigating all possible technical avenues to solving QCS. In particular, we do not investigate fine-tuning sequence-to-sequence models as preliminary experiments showed poor performance. In addition, we do not explore the full range of abilities offered to us by LLMs such as GPT-3.5. For instance, using GPT-3.5 to generate synthetic annotations may have helped the performance of our larger neural-based models which are known to perform poorly in data-scarce settings.

## Ethics Statement

We strongly believe in the work done by qualitative researchers and the *human* understanding that they provide to complex bodies of text. This work is an effort to develop methods that *help* rather than replace the arduous work conducted by qualitative researchers and coders.

The data collection of this study has been approved by the Faculty of Medicine and Health Sciences Institutional Review Board (IRB) of McGill University (IRB Internal Study Number: A03-B22-21A). All necessary participant consent has been obtained and the appropriate institutional forms have been archived.

## Acknowledgements

We would like to thank the anonymous reviewers for their feedback and valuable suggestions. In addition, we would like to thank Andrei Mircea, Aylin Erman, Dan Poenaru, Jad Kabbara, Ian Porada, Ines Arous, Sabina Elkins and Yu Lu Liu for their insightful feedback on the multiple iterations of this paper.

This work was supported by the Fonds de Recherche du Québec – Nature et Technologies (FRQNT). Jackie Chi Kit Cheung is supported by the Canada CIFAR AI Chair program. Samira Abbasgholizadeh Rahimi is a Canada Research Chair (Tier 2) in Advanced Digital Primary Health Care, and received salary support (Research Scholar Junior 1 Career Development Award) from the Fonds de Recherche du Québec-Santé (FRQS) during this study.

The authors acknowledge the material support of NVIDIA in the form of computational resources.

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

# A Appendix

## A.1 Dataset Characteristics

| Source | Corpus size (in number of words) | Number of annotations | Publicly available? |
|---|---|---|---|
| (Crowston et al., 2012) | 84870 | 3011 | No |
| (Grandeit et al., 2020) | N/A | 10000 | No |
| (Xiao et al., 2023) | N/A | 668 | No |
| CVDQuoding | 63927 | 1175 | Yes[7] |

Table 4: Comparative analysis of our dataset with previously closed-source datasets. In this case, the number of words are counted using space separation and the number of annotations refer to the number of text spans that have been assigned one or more codes.

```
<question_2>Interviewer: Have you ever been informed about your cardiovascular
    health, and if so how have you been informed?</question_2>

Participant 1: <code coder='2' value='Understanding of CVD'>So, I've been in
    cardiology since '96. I used to work at the Montreal Heart Institute before so
    <code coder='1' value='Sources of information'>I attended cardiac rounds and
    went to conferences and things like that. I also like to read on health, so
    media, papers and then conferences.
```

Figure 6: Excerpt from Interview 1 of the CVDQuoding dataset with XML tags to identify the question asked and the text spans that have been annotated. In the code tag, the attribute coder identifies which qualitative coder was responsible for the annotation and the attribute value identifies the code assigned to the text span.

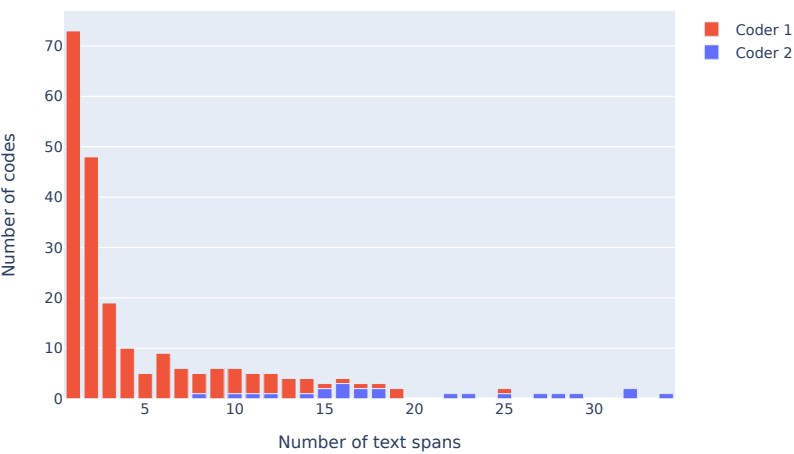

Figure 7: Frequency of code assignment counts.

[7]Email the corresponding author.

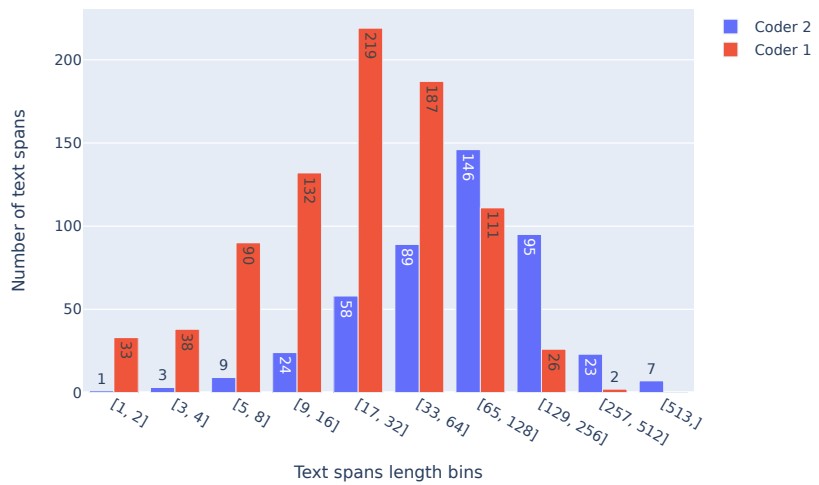

Figure 8: Distribution of the length of text spans.

| Annotator | List of codes |
|---|---|
| Coder 1 | Life-style modification, Needs to assure data confidentiality and security, Increasing trust in the tool, Lack of knowledge, Additional comments about diet recommendations, CVD misconceptions, Level of interest in tracking information and prompts, Level of interest in educational modules, Additional comments about educational modules, Level of interest in transparency and explainability, Enthusiastic about technology, App prompts, Frequency of push-notifications, Level of interest in personalized workouts, Additional comments about explainability |
| Coder 2 | Technology for CVD care, Cardiovascular Health Information, Accessibility, Decisions Made, Additional Features Suggested, Understanding of CVD, Challenges, Trust and Reliability, Transparency, Decision Involvement, Weight Tracking, Educational Models, Technical Support, Data Monitoring, Pedometer |

Table 5: List of 15 codes created by coders 1 and 2 sorted by frequency over the entire `CVDQuoding` dataset.

## A.2 Interview Questions

| | Interview Questions |
|---|---|
| Question 1 | How do you think cardiovascular diseases are generally described and understood by the public? |
| Question 2 | Have you ever been informed about your cardiovascular health? How? |
| Question 3 | What are the most frequent and important decisions you face related to your cardio-vascular health? |
| Question 4 | What is your usual role in making decisions about your cardiovascular health and preventing CVD? |
| Question 5 | What do you think are some challenges and needs in preventing and managing cardiovascular diseases (from your perspective as a woman at risk of CVD)? |
| Question 6 | Have you ever considered a decision support system to help you in any decisions related to your cardiovascular health? |
| Question 7 | What are your thoughts on using digital technology (e.g., mobile apps, AI systems/robots) to make decisions in relation to your cardiovascular health? |
| Question 8 | How would you like us to design and develop this Xi-Care tool that is useful, helpful and effective for women at risk of CVD (e.g. no risks to users)? |
| Question 9 | On a scale of 1 to 5 with 1 being not at all interested and 5 being very interested, what is your level of interest in monitoring tools that track health data over time? |
| Question 10 | On a scale of 1 to 5 with 1 being not at all interested and 5 being very interested, what is your level of interest in a step-count feature? |
| Question 11 | On a scale of 1 to 5 with 1 being not at all interested and 5 being very interested, what is your level of interest in a weight tracking feature? |
| Question 12 | On a scale of 1 to 5 with 1 being not at all interested and 5 being very interested, what is your level of interest in educational modules on cardiovascular health? |
| Question 13 | On a scale of 1 to 5 with 1 being not at all interested and 5 being very interested, what is your level of interest in guided exercise activities? |
| Question 14 | On a scale of 1 to 5 with 1 being not at all interested and 5 being very interested, what is your level of interest in diet recommendations? |
| Question 15 | Would you like to be able to follow your progress and receive push-notifications through the Xi-Care tool? |
| Question 16 | How difficult/easy do you think it will be for you to integrate the Xi-Care tool into your daily life? |
| Question 17 | To what extent would you trust or rely on the Xi-Care tool to make assessments about your cardiovascular health and prevent and manage CVD? |
| Question 18 | Do you foresee any challenges with integrating the Xi-Care tool in your daily life in terms of ethics? Could you please describe these challenges? |
| Question 19 | In terms of transparency, how important is it for you to be able to understand how the Xi-Care tool works? |
| Question 20 | Is there something else you'd like to add about ethical aspects in regards to the Xi-Care tool that will be empowered by AI (Justice; Non-maleficence; Autonomy; Beneficence; Explicability/Transparency)? |

Table 6: List of questions asked during the interviews of the `CVDQuoding` dataset. *Xi-Care* is the name of the app being proposed to participants of the study to help them control their risks of cardiovascular diseases (CVDs).

### A.3 Hyperparameter Configurations

We provide additional hyperparameter and architecture details for all our models.

#### A.3.1 Baseline

**Okapi-BM25.** We use an open-source implementation of the BM25 algorithm[8]. We use the Okapi implementation with its default parameters $k_1 = 1.5, b = 0.75, \varepsilon = 0.25$ and a tokenizer with space separation and lowercasing.

#### A.3.2 Classification Paradigm

**SVM.** We use the SVM implementation from sklearn[9]. We transform all inputs to tf-idf features with separate encodings for the question and text span. We use the standard radial basis function kernel and tune the regularization parameter $C$ using the values $\{0.001, 0.01, 0.1, 1, 10\}$. A class weight parameter is computed for each code as well as for the novel code and is inversely proportional to the class frequencies in the training set.

**DistilBERT.** We use the DistilBERT implementation from Hugging Face[10]. In particular, we use DistilBERT's tokenizer and its DistilBertForSequenceClassification module which accepts a variable number of labels, $|C| + 1$ in this case. We tune the learning rate using values $\{1 \times 10^{-5}, 2 \times 10^{-5}\}$ with a weight decay of $0.01$. We use a batch size of 8 and train for 25 epochs with early stopping on the validation loss with a patience of 5 epochs. We also compute a class weight inversely proportional to the class frequencies in the training set and apply it to the cross entropy loss. All our experiments consistently show that using class weights benefits performance.

#### A.3.3 Information-retrieval Paradigm

Along with hyperparameter details, we provide additional architecture details about both the bi-encoder and cross-encoder.

**Bi-Encoder.** In the bi-encoder, we create representations for the code $c$, the text span $t$ and the previous question $q$ with three distinct BERT encoders, $E_t$, $E_q$ and $E_c$. In our case, we use DistilBERT as the encoder for all three inputs. We pass each input whose start is truncated to the model's 512 token input limit to its respective encoder and extract its CLS token representation.

$$h_t = E_t(t), h_q = E_q(q), h_c = E_c(c)$$

We use a component-wise max pooling method $\max(\cdot, \cdot)$ to aggregate $h_t$ and $h_q$ together.

$$h_{\text{pool}} = \max(h_t, h_q)$$

Finally, a cosine similarity is applied between the pooled representation, $h_{\text{pool}}$, and the code representation, $h_c$, to generate a score. A *class weighted* mean-squared error loss is then computed and backpropagated through the three encoders. For the novel code detection subtask, a linear classifier is placed on top of the vector of encoder scores. For the training of the linear classifier, we use another class weighted cross entropy loss and keep the three encoders frozen.

We tune the learning rate using the values $\{1 \times 10^{-5}, 2 \times 10^{-5}\}$ with a weight decay of $0.01$. We use a larger batch size of 32, since training sets now have a size of $|C| \times N$ where $N$ is the number of coded text spans. We train for 25 epochs with early stopping on the validation loss set with a patience of 5 epochs.

**Cross-Encoder.** In the cross-encoder, we create representations for the concatenation of codes with the question and the text span. That is, for every code $c$ in the training set, the text span $t$ and the previous question $q$ are each independently concatenated with the code $c$ using the special [SEP] token. Both concatenations are passed to different encoders $E_t$ and $E_q$ to obtain a contextual representation. Truncation is applied to the start of both $t$ and $q$ to comply with the encoder's maximum token input length.

$$h_{t \, [\text{SEP}] \, c} = E_t(t \, [\text{SEP}] \, c)$$
$$h_{q \, [\text{SEP}] \, c} = E_q(q \, [\text{SEP}] \, c)$$

The representations are pooled using a component-wise max pooling method $\max(\cdot, \cdot)$

$$h_{\text{pool}} = \max(h_{t \, [\text{SEP}] \, c}, h_{q \, [\text{SEP}] \, c})$$

We place a classification head on top of $h_{\text{pool}}$ identical in architecture to the DistilBERT classification head. We compute a *class weighted* cross entropy loss and backpropagate it through both encoders. For the novel code detection subtask, a linear classifier is placed on top of the vector of encoder scores, which, in this case, are logits. We use a class weighted cross entropy loss and keep the two encoders frozen.

In this case, we experiment with DistilBERT and ConvBERT. In both cases, we tune the learning

---

[8] https://github.com/dorianbrown/rank_bm25
[9] https://scikit-learn.org/stable/
[10] https://huggingface.co/

rate using the values $\{1 \times 10^{-5}, 2 \times 10^{-5}\}$ with a weight decay of $0.01$. We use a batch size of $32$ and train for $30$ epochs with early stopping on the validation loss set to have a patience of $5$ epochs. We train for more epochs because we noticed that the cross-encoders took longer to converge.

### A.3.4   Zero-shot Paradigm

We use OpenAI's GPT-3.5 API to generate LLM responses to our zero-shot prompt. In particular, we use the model checkpoint `gpt-3.5-turbo-0301` with a temperature of $1$. The question and the text span are truncated to allow for a $64$ token generation. At the time of our experiments, this version of GPT-3.5 allowed for $4096$ tokens. In Figure 9, we show the template that we use to prompt GPT-3.5.

> You are a helpful assistant that suggests qualitative codes for a qualitative researcher. The coders you can suggest are: [AVAILABLE_CODES], and None of the above. Which of the previous codes would you assign to the following excerpt from an interview with a woman at risk of cardiovascular disease (CVD): "Question: [QUESTION] Answer: [ANSWER]"

Figure 9: Template used to prompt a generative LLM $M$ for a ranking of the codes for a passage. [AVAILABLE_CODES] is a placeholder for the list of codes from $\mathcal{C}$ to rank for an instance $(q, t)_i$, [QUESTION] is a placeholder for the previous question $q$ and [ANSWER] is a placeholder for the text span $t$ highlighted by one of the coders.

## A.4 Automatic Metrics

We provide complete descriptions for the automatic metrics presented in Section 6.2.1 and 6.2.2.

**MRR**  To compute the MRR, we consider the set of true codes, $C_i^{\text{true}}$ assigned to a text span $i$ as well as the predicted rank, $\texttt{predicted\_rank}_i : \mathcal{C} \to \{1, \ldots, |C|\}$, assigned to each code $c \in \mathcal{C}$ for text span $i$ by one of our systems. In this computation, we exclude the $\texttt{novel}$ code as we are interested in a system's ability to suggest previously assigned codes. The reciprocal rank, $\text{RR}_i$, for a passage $i$ is computed as the maximum predicted reciprocal rank across all true codes. That is,

$$\text{RR}_i = \max_{c \in C_i^{\text{true}}} \frac{1}{\texttt{predicted\_rank}_i(c)}$$

The MRR is then computed by averaging across all test instances $i$ with $|C_i^{\text{true}}| \geq 1$.

**P@k**  To compute the P@k, let $N_k$ be the number of relevant codes, as indicated by some gold standard annotation, in the first $k$ suggested codes. Then the P@k for a text span $i$ is computed as

$$\text{P@k}_i = \frac{N_k}{k}$$

We then average the instance-level P@k over all test instances.

**sNDCG@k**  To compute the sNDCG metric, we approximate a suggested code's $c$ relevance score for a text span $i$, $rel_i(c)$, in order to carry out the standard NDCG computation. The relevance score, $rel_i(c)$, is computed by using the BERTScore (Zhang et al., 2019) as the approximation for the affinity between suggested and true codes. More specifically, the relevance score $rel_i(c)$, for a code $c \in \mathcal{C}$ and a text span $i$ is

$$rel_i(c) = \max_{c' \in C_i^{\text{true}}} \text{BERTScore}(c, c')$$

Once all the relevance scores are computed, we use them in the standard NDCG computation. To do so, we sort the list of relevance factors $(rel_i(c) : c \in \mathcal{C})$ to compute the $\texttt{true\_rank\_scores}$ list for a text span $i$. In addition, the $\texttt{predicted\_rank\_scores}$ for a text span $i$ is computed as the list of relevance factors sorted by the model's scores for each code. We can then compute the sNDCG score for the suggestions made by our system for text span $i$ as

$$\text{sNDCG}_i = \frac{\text{sDCG}_i}{\text{sIDCG}_i}$$

$$\text{sIDCG}_i = \sum_{i \in 2\ldots|\mathcal{C}|+1} \frac{\texttt{true\_rank\_scores}[i]}{\log(i)}$$

$$\text{sDCG}_i = \sum_{i \in 2\ldots|\mathcal{C}|+1} \frac{\texttt{predicted\_rank\_scores}[i]}{\log(i)}$$

Finally, we add a cutoff $k$ to sNDCG as is usually done in standard NDCG to account only for the top $k$ results creating sNDCG@$k$.

### A.5 Human Evaluation

We conducted the human evaluation by first running a pilot study. The pilot study was used to assess whether the instructions of the human evaluation were clear and to estimate the duration of the study for the full 32 samples. During the pilot study, we recruited a researcher familiar with qualitative research and asked them to judge the relevance of 8 instances. Half of these 8 instances were sampled from coder 1's human evaluation samples, and the other half came from coder 2's. We made sure that at least one code had been assigned to every passage we showed evaluators as we wanted to be able to compare the rank-based metrics computed using the original annotations and the human evaluation annotations. In addition, we always included the codes from the original coding annotations to avoid having the evaluator mark all suggestions as irrelevant. An example of a question and its corresponding highlighted text span from coder 1's annotations as well as a list of codes (truncated due to space constraints) used to gather human relevance judgements is shown in Figure 10.

Question: So my next question is what are your thoughts on using digital technology, for example, mobile apps, AI systems, to make decisions as they relate to your cardiovascular health or health, in general?

Answer: I think it's more effective. I can also manage it and I can show it to my doctor. Doctors, they don't have enough time to listen. Maybe if I show them with this app, the doctor can clearly see this year you got this, this year you got this. I think it's also helpful for health professionals, along with helping regular patients.

| | Relevant | Irrelevant |
|---|---|---|
| Professional consultation | ● | ○ |
| Preference of tool being supported by different devices | ○ | ● |
| Not being aware of AI utilization | ○ | ● |
| Sources of information | ○ | ● |
| Independent research of patients | ○ | ● |
| Benefits of automatic data collection | ● | ○ |

Figure 10: An example of a question and a text span from coder 1's annotations shown to the human evaluator. We truncate the list of codes shown due to space constraints.

## A.6 Additional Results

### A.6.1 sNDCG Plots

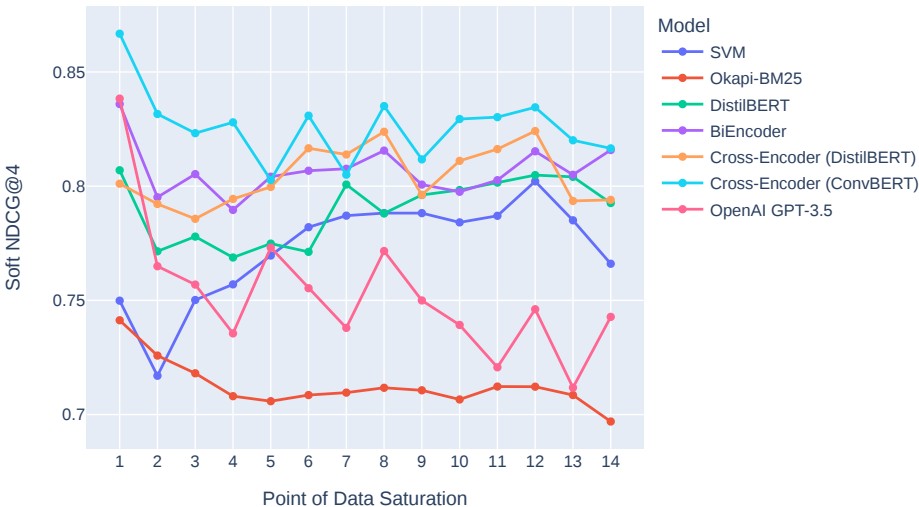

Figure 11: Plot of sNDCG@4 for coder 1 across $K = 1$ to $K = 14$

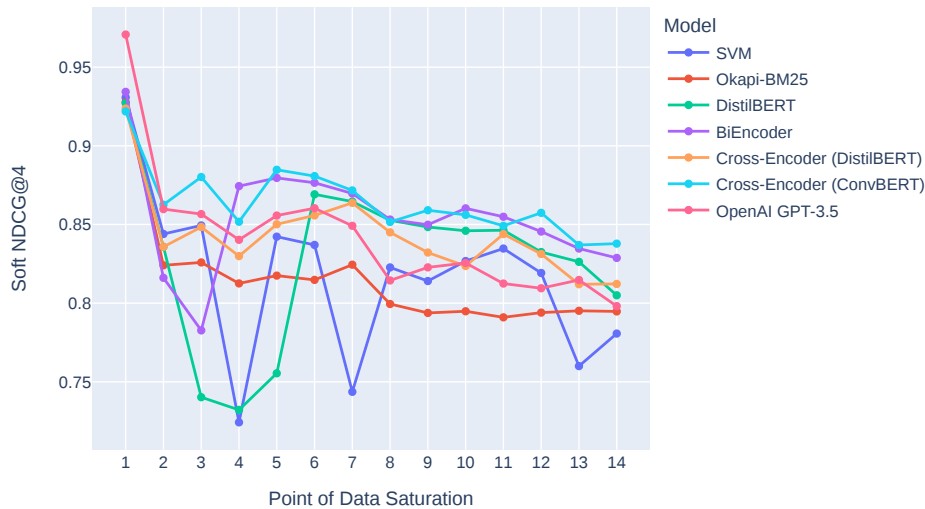

Figure 12: Plot of sNDCG@4 for coder 2 across $K = 1$ to $K = 14$

## A.6.2 MRR Tables

| | 1 | 2 | 3 | 4 | 5 | 6 | 7 | 8 | 9 | 10 | 11 | 12 | 13 | 14 |
|---|---|---|---|---|---|---|---|---|---|---|---|---|---|---|
| SVM | 0.28 | 0.29 | 0.39 | 0.41 | 0.44 | 0.49 | 0.52 | 0.5 | 0.52 | 0.51 | 0.51 | 0.54 | 0.47 | 0.42 |
| DistilBERT | 0.45 | 0.39 | 0.42 | 0.43 | 0.43 | 0.48 | 0.52 | 0.5 | 0.55 | 0.55 | 0.57 | 0.61 | 0.57 | 0.57 |
| Okapi-BM25 | 0.19 | 0.19 | 0.17 | 0.14 | 0.18 | 0.14 | 0.17 | 0.17 | 0.14 | 0.15 | 0.14 | 0.11 | 0.18 | 0.17 |
| Bi-Encoder | 0.55 | 0.41 | 0.44 | 0.4 | 0.45 | 0.48 | 0.51 | 0.54 | 0.48 | 0.48 | 0.54 | 0.54 | 0.54 | 0.55 |
| Cross-Encoder (DistilBERT) | 0.41 | 0.45 | 0.39 | 0.51 | 0.5 | 0.55 | 0.57 | 0.56 | 0.55 | 0.55 | 0.56 | 0.61 | 0.56 | 0.54 |
| Cross-Encoder (ConvBERT) | 0.69 | 0.59 | 0.52 | 0.56 | 0.42 | 0.63 | 0.58 | **0.61** | 0.56 | **0.59** | **0.60** | 0.62 | 0.57 | 0.52 |
| GPT-3.5 | **0.74** | **0.69** | **0.66** | **0.61** | **0.64** | **0.66** | **0.58** | 0.6 | **0.63** | 0.57 | 0.56 | **0.63** | **0.58** | **0.58** |

Table 7: MRR for coder 1 for $K = 1$ to $K = 14$

| | 1 | 2 | 3 | 4 | 5 | 6 | 7 | 8 | 9 | 10 | 11 | 12 | 13 | 14 |
|---|---|---|---|---|---|---|---|---|---|---|---|---|---|---|
| SVM | 0.65 | 0.63 | 0.63 | 0.27 | 0.67 | 0.64 | 0.36 | 0.66 | 0.62 | 0.68 | 0.7 | 0.65 | 0.46 | 0.48 |
| DistilBERT | 0.67 | 0.6 | 0.25 | 0.27 | 0.35 | 0.76 | **0.76** | **0.75** | 0.75 | 0.75 | 0.74 | 0.7 | 0.71 | 0.63 |
| Okapi-BM25 | 0.23 | 0.28 | 0.23 | 0.22 | 0.23 | 0.23 | 0.22 | 0.24 | 0.23 | 0.23 | 0.23 | 0.23 | 0.2 | 0.19 |
| Bi-Encoder | 0.67 | 0.5 | 0.41 | 0.73 | 0.76 | 0.74 | 0.7 | 0.71 | 0.7 | 0.74 | 0.76 | 0.73 | 0.71 | 0.68 |
| Cross-Encoder (DistilBERT) | 0.62 | 0.62 | 0.63 | 0.64 | 0.67 | 0.69 | 0.72 | 0.72 | 0.68 | 0.64 | 0.71 | 0.69 | 0.64 | 0.59 |
| Cross-Encoder (ConvBERT) | 0.62 | 0.7 | 0.74 | 0.67 | **0.78** | **0.78** | 0.76 | 0.75 | **0.77** | **0.77** | **0.78** | **0.76** | **0.72** | 0.72 |
| GPT-3.5 | **0.89** | **0.78** | **0.76** | **0.74** | 0.77 | 0.77 | 0.76 | 0.74 | 0.73 | 0.73 | 0.73 | 0.71 | 0.71 | **0.75** |

Table 8: MRR for coder 2 for $K = 1$ to $K = 14$

## A.6.3 sNDCG@4 Tables

| | 1 | 2 | 3 | 4 | 5 | 6 | 7 | 8 | 9 | 10 | 11 | 12 | 13 | 14 |
|---|---|---|---|---|---|---|---|---|---|---|---|---|---|---|
| SVM | 0.75 | 0.72 | 0.75 | 0.76 | 0.77 | 0.78 | 0.79 | 0.79 | 0.79 | 0.78 | 0.79 | 0.8 | 0.79 | 0.77 |
| DistilBERT | 0.81 | 0.77 | 0.78 | 0.77 | 0.77 | 0.77 | 0.8 | 0.79 | 0.8 | 0.8 | 0.8 | 0.8 | 0.8 | 0.79 |
| Okapi-BM25 | 0.74 | 0.73 | 0.72 | 0.71 | 0.71 | 0.71 | 0.71 | 0.71 | 0.71 | 0.71 | 0.71 | 0.71 | 0.71 | 0.7 |
| Bi-Encoder | 0.84 | 0.8 | 0.81 | 0.79 | **0.80** | 0.81 | 0.81 | 0.82 | 0.8 | 0.8 | 0.8 | 0.82 | 0.8 | 0.82 |
| Cross-Encoder (DistilBERT) | 0.8 | 0.79 | 0.79 | 0.79 | 0.8 | 0.82 | **0.81** | 0.82 | 0.8 | 0.81 | 0.82 | 0.82 | 0.79 | 0.79 |
| Cross-Encoder (ConvBERT) | **0.87** | **0.83** | **0.82** | **0.83** | 0.8 | **0.83** | 0.81 | **0.84** | **0.81** | **0.83** | **0.83** | **0.83** | **0.82** | **0.82** |
| GPT-3.5 | 0.84 | 0.76 | 0.76 | 0.74 | 0.77 | 0.76 | 0.74 | 0.77 | 0.75 | 0.74 | 0.72 | 0.75 | 0.71 | 0.74 |

Table 9: sNDCG@4 for coder 1 for $K = 1$ to $K = 14$

| | 1 | 2 | 3 | 4 | 5 | 6 | 7 | 8 | 9 | 10 | 11 | 12 | 13 | 14 |
|---|---|---|---|---|---|---|---|---|---|---|---|---|---|---|
| SVM | 0.93 | 0.84 | 0.85 | 0.72 | 0.84 | 0.84 | 0.74 | 0.82 | 0.81 | 0.83 | 0.83 | 0.82 | 0.76 | 0.78 |
| DistilBERT | 0.93 | 0.84 | 0.74 | 0.73 | 0.76 | 0.87 | 0.86 | 0.85 | 0.85 | 0.85 | 0.85 | 0.83 | 0.83 | 0.8 |
| Okapi-BM25 | 0.93 | 0.82 | 0.83 | 0.81 | 0.82 | 0.81 | 0.82 | 0.8 | 0.79 | 0.79 | 0.79 | 0.79 | 0.8 | 0.79 |
| Bi-Encoder | 0.93 | 0.82 | 0.78 | **0.87** | 0.88 | 0.88 | 0.87 | **0.85** | 0.85 | **0.86** | **0.85** | 0.85 | 0.83 | 0.83 |
| Cross-Encoder (DistilBERT) | 0.92 | 0.84 | 0.85 | 0.83 | 0.85 | 0.86 | 0.86 | 0.84 | 0.83 | 0.82 | 0.84 | 0.83 | 0.81 | 0.81 |
| Cross-Encoder (ConvBERT) | 0.92 | **0.86** | **0.88** | 0.85 | **0.88** | **0.88** | **0.87** | 0.85 | **0.86** | 0.86 | 0.85 | **0.86** | **0.84** | **0.84** |
| GPT-3.5 | **0.97** | 0.86 | 0.86 | 0.84 | 0.86 | 0.86 | 0.85 | 0.81 | 0.82 | 0.83 | 0.81 | 0.81 | 0.81 | 0.8 |

Table 10: sNDCG@4 for coder 2 for $K = 1$ to $K = 14$

### A.6.4  Macro $F_1$ Tables

| | 1 | 2 | 3 | 4 | 5 | 6 | 7 | 8 | 9 | 10 | 11 | 12 | 13 | 14 |
|---|---|---|---|---|---|---|---|---|---|---|---|---|---|---|
| SVM | 0.44 | 0.37 | 0.37 | 0.33 | 0.32 | 0.33 | 0.35 | 0.35 | 0.34 | 0.35 | 0.33 | 0.32 | 0.3 | 0.21 |
| DistilBERT | 0.49 | 0.52 | **0.58** | 0.56 | 0.55 | **0.62** | **0.60** | **0.63** | **0.61** | **0.62** | **0.64** | **0.65** | **0.63** | **0.7** |
| Okapi-BM25 | 0.48 | 0.5 | 0.56 | **0.56** | **0.56** | 0.54 | 0.57 | 0.49 | 0.56 | 0.55 | 0.56 | 0.56 | 0.53 | 0.58 |
| BiEncoder | 0.36 | 0.3 | 0.32 | 0.38 | 0.36 | 0.38 | 0.39 | 0.4 | 0.41 | 0.42 | 0.42 | 0.43 | 0.44 | 0.46 |
| Cross-Encoder (DistilBERT) | **0.53** | **0.53** | 0.35 | 0.51 | 0.51 | 0.55 | 0.6 | 0.45 | 0.58 | 0.59 | 0.59 | 0.49 | 0.55 | 0.53 |
| Cross-Encoder (ConvBERT) | 0.49 | 0.48 | 0.38 | 0.5 | 0.36 | 0.5 | 0.41 | 0.56 | 0.59 | 0.47 | 0.55 | 0.61 | 0.59 | 0.51 |
| OpenAI GPT-3.5 (Turbo) | 0.26 | 0.37 | 0.4 | 0.46 | 0.41 | 0.42 | 0.42 | 0.44 | 0.47 | 0.53 | 0.44 | 0.52 | 0.41 | 0.44 |

Table 11: Macro F1 for the novel code for coder 1 for $K = 1$ to $K = 14$

| | 1 | 2 | 3 | 4 | 5 | 6 | 7 | 8 | 9 | 10 | 11 | 12 | 13 | 14 |
|---|---|---|---|---|---|---|---|---|---|---|---|---|---|---|
| SVM | 0.1 | 0.27 | 0.29 | 0.47 | 0.3 | 0.31 | 0.17 | - | - | - | - | - | - | - |
| DistilBERT | 0.49 | 0.49 | 0.4 | 0.43 | 0.45 | 0.45 | 0.45 | - | - | - | - | - | - | - |
| Okapi-BM25 | 0.44 | 0.6 | 0.56 | **0.50** | 0.53 | 0.5 | 0.56 | - | - | - | - | - | - | - |
| BiEncoder | **0.52** | 0.39 | 0.4 | 0.47 | 0.45 | 0.45 | 0.45 | - | - | - | - | - | - | - |
| Cross-Encoder (DistilBERT) | 0.47 | **0.63** | 0.56 | 0.46 | 0.46 | **0.56** | 0.59 | - | - | - | - | - | - | - |
| Cross-Encoder (ConvBERT) | 0.47 | 0.56 | **0.64** | 0.47 | **0.57** | 0.55 | 0.57 | - | - | - | - | - | - | - |
| OpenAI GPT-3.5 (Turbo) | 0.13 | 0.42 | 0.42 | 0.49 | 0.45 | 0.46 | 0.48 | - | - | - | - | - | - | - |

Table 12: Macro F1 for the novel code prediction subtask for coder 2 for $K = 1$ to $K = 14$. Cells past $K = 8$ are marked with "-" since no novel codes occur.

### A.6.5  Micro $F_1$ Tables

| | 1 | 2 | 3 | 4 | 5 | 6 | 7 | 8 | 9 | 10 | 11 | 12 | 13 | 14 |
|---|---|---|---|---|---|---|---|---|---|---|---|---|---|---|
| SVM | **0.79** | **0.58** | 0.56 | 0.46 | 0.44 | 0.4 | 0.41 | 0.39 | 0.38 | 0.37 | 0.35 | 0.34 | 0.31 | 0.22 |
| DistilBERT | 0.51 | 0.53 | **0.58** | **0.60** | 0.59 | **0.66** | **0.65** | **0.68** | 0.67 | 0.71 | 0.69 | **0.74** | 0.75 | 0.84 |
| Okapi-BM25 | 0.55 | 0.53 | 0.57 | 0.57 | 0.59 | 0.63 | 0.65 | 0.6 | 0.63 | 0.62 | 0.63 | 0.64 | 0.64 | 0.76 |
| BiEncoder | 0.36 | 0.42 | 0.44 | 0.56 | 0.57 | 0.62 | 0.63 | 0.68 | 0.69 | **0.72** | **0.73** | 0.74 | 0.77 | **0.86** |
| Cross-Encoder (DistilBERT) | 0.56 | 0.53 | 0.45 | 0.54 | **0.59** | 0.61 | 0.64 | 0.68 | 0.68 | 0.64 | 0.64 | 0.7 | **0.8** | 0.79 |
| Cross-Encoder (ConvBERT) | 0.74 | 0.53 | 0.53 | 0.51 | 0.57 | 0.63 | 0.62 | 0.56 | **0.71** | 0.7 | 0.66 | 0.64 | 0.62 | 0.61 |
| OpenAI GPT-3.5 (Turbo) | 0.28 | 0.44 | 0.47 | 0.58 | 0.55 | 0.57 | 0.6 | 0.64 | 0.66 | 0.71 | 0.67 | 0.71 | 0.69 | 0.78 |

Table 13: Micro F1 for the novel code prediction subtask for coder 1 for $K = 1$ to $K = 14$.

| | 1 | 2 | 3 | 4 | 5 | 6 | 7 | 8 | 9 | 10 | 11 | 12 | 13 | 14 |
|---|---|---|---|---|---|---|---|---|---|---|---|---|---|---|
| SVM | 0.12 | 0.36 | 0.36 | **0.87** | 0.3 | 0.32 | 0.19 | - | - | - | - | - | - | - |
| DistilBERT | 0.62 | 0.58 | 0.66 | 0.56 | **0.80** | **0.81** | 0.83 | - | - | - | - | - | - | - |
| Okapi-BM25 | 0.63 | 0.62 | 0.58 | 0.71 | 0.76 | 0.68 | 0.79 | - | - | - | - | - | - | - |
| BiEncoder | 0.77 | 0.64 | 0.66 | 0.87 | 0.8 | 0.81 | **0.83** | - | - | - | - | - | - | - |
| Cross-Encoder (DistilBERT) | **0.88** | **0.67** | 0.68 | 0.87 | 0.78 | 0.74 | 0.8 | - | - | - | - | - | - | - |
| Cross-Encoder (ConvBERT) | 0.88 | 0.65 | **0.70** | 0.87 | 0.79 | 0.76 | 0.74 | - | - | - | - | - | - | - |
| OpenAI GPT-3.5 (Turbo) | 0.14 | 0.65 | 0.65 | 0.87 | 0.8 | 0.79 | 0.83 | - | - | - | - | - | - | - |

Table 14: Micro F1 for the novel code prediction subtask for coder 2 for $K = 1$ to $K = 14$. Cells past $K = 8$ are marked with "-" since no novel codes occur.