# OpenReview forum: "Qualitative Code Suggestion: A Human-Centric Approach to Qualitative Coding"
_EMNLP/2023/Conference — EMNLP 2023 Findings_

### Official Review · Reviewer_ULRd · 2023-08-02

**Soundness:** 4

**Excitement:**

4: Strong: This paper deepens the understanding of some phenomenon or lowers the barriers to an existing research direction.

**Missing References:**

To me it seems like this proposed task largely overlaps to a large degree with human-in-the-loop annotation, which is a well-studied topic. Tools exist, like Prodigy for instance (Montani & Honnibal, 2018). And also previous ACL-research, such as:

Yu et al. (2015). LSUN: Construction of a large-scale image dataset using deep learning with humans in the loop
Liu et al. (2019). Deep reinforcement active learning for human-in-the-loop person re-identification.
Fan et al. (2019). An interactive visual analytics approach for network anomaly detection through smart labeling.
Gesller et al. (2022). Midas Loop: A Prioritized Human-in-the-Loop Annotation for Large Scale Multilayer Data.
Mendes et al. (2023). Human-in-the-loop Evaluation for Early Misinformation Detection: A Case Study of COVID-19 Treatments.
Klie et al. (2020). From Zero to Hero: Human-In-The-Loop Entity Linking in Low Resource Domains.

It would have been nice if this work had been discussed in this paper to compare previous approaches on this topic to their approach, and maybe also to provide relevant baselines to compare their QCS to.

**Paper Topic And Main Contributions:**

This paper describes a novel approach to perform qualitative annotation. Previous research mostly aimed to create a system that does the annotation process fully automatically, while this paper aims to provide "qualitative code suggestion" so that the annotation process becomes a collaboration between human and machine. Furthermore, the authors aim for their approach to not only suggest pre-defined codes, but also detect and suggest novel, rare, codes that have not been defined. It should also be mentioned that the authors introduce the Quoding dataset, which is the only open-source dataset for this type of task.

The results show that their human-centric approach to qualitative coding can be of great assistance to annotators. However, the system is not capable of performing the novel code detection subtask.

**Questions For The Authors:**

- §2 (On deductive vs. inductive coding) --> What would make inductive coding fundamentally different? I would think that inductive coding just adds an extra step of exploratory analysis where codes are defined before moving on to the official annotation. However, this annotation itself would then not be different from deductive coding.

**Reasons To Accept:**

- Potentially helpful tool to many researchers that perform qualitative coding in some capacity
- Introduction of an open-source dataset for this task
- Extensive analysis of the results (section 8 is quite valuable)

**Reasons To Reject:**

While the authors do a good job of reporting and describing their results, I do have some questions related to clarity and methodological choices.

- Questionable to what degree this task is novel (therefore missing relevant baselines) (see "Missing References").
- Evaluation does not seem most relevant for this task. The main purpose of this system is to help annotators do their work more efficiently. Do the suggestions make annotation more efficient (in terms of annotation time), and do they decrease Annotation Errors by the annotators when used, for instance?

**Reproducibility:**

4: Could mostly reproduce the results, but there may be some variation because of sample variance or minor variations in their interpretation of the protocol or method.

**Reviewer Confidence:**

4: Quite sure. I tried to check the important points carefully. It's unlikely, though conceivable, that I missed something that should affect my ratings.

---

> ### Author Rebuttal · Authors · 2023-08-27
>
> We would like to thank the reviewer for their comments and feedback. In particular, we appreciate the reviewer highlighting the potential of our QCS systems to become full-fledged tools which could assist qualitative coders in their annotations. We also appreciate the reviewer pointing out the development of the Quoding dataset as well as the level of detail and the value of the analysis of our results (Section 8).
>
> We respond to the reviewer’s main points and answer their questions below:
>
> Firstly, the difference between inductive and deductive coding lies in the fact that each coding technique serves a significantly different purpose. Deductive coding is used to validate or invalidate a pre-defined hypothesis about a corpus. This purpose is why, in deductive coding, researchers design a codebook based on a research question about a corpus and then code the corpus looking specifically for text spans supporting those codes. In contrast, inductive coding is used to discover new phenomena underlying a corpus. This purpose is why, in inductive coding, researchers do not have a pre-defined research question, a codebook is not designed prior to coding and prior exploratory data analysis is not recommended. This difference is one of the reasons why existing fully automatic coding systems (which frame coding as text classification) are not suitable for inductive coding and is what motivated the creation of the QCS task.
>
> Secondly, there is a fundamental difference between qualitative coding and other annotation processes which human-in-the-loop papers have studied. Not only is qualitative coding an annotation process, but it is also - and more importantly - a way for researchers to understand and familiarize themselves with their data. As Jiang et al. (2021) point out in their paper on the challenges of human-AI techniques for qualitative coding, “qualitative analysis is a unique case for human-AI collaboration because uncertainty is essential to qualitative analysis itself.” In other words, qualitative coding is unique because the act of coding data, and the uncertainty that comes with it, is valued by researchers as it cultivates their understanding of their corpus. We would be happy to add additional references to the camera-ready version of the paper to explain how the nature of qualitative coding is fundamentally different from that of other annotation methods.
>
> Finally, as discussed in a response to another reviewer, the human evaluation we perform is not meant to be seen as a user study which investigates how the qualitative coders react and work with a QCS method. The primary purpose of the human evaluation is to validate the relevance of the code suggestions made by each QCS method. As we discuss in the Results section (7.2), our human evaluation shows that the most performant QCS systems (ConvBERT and GPT-3.5) are able to consistently make relevant code suggestions. We have left the study of how relevant code suggestions impact the workflow and findings of qualitative coding for future work as we believe this separate endeavour merits its own human-computer interaction publication. We would be happy to clarify the purpose of the human evaluation and its results in the camera-ready version of the paper. In addition, we would be happy to discuss the necessity of a comprehensive user study in the Conclusion as future work.
>
> Thanks again to the reviewer for their constructive feedback and useful comments.

---

### Official Review · Reviewer_WMQc · 2023-08-05

**Soundness:** 4

**Excitement:**

4: Strong: This paper deepens the understanding of some phenomenon or lowers the barriers to an existing research direction.

**Paper Topic And Main Contributions:**

The paper proposes a user-centric approach to assist qualitative researchers by providing automated agent generated code suggestions. The key contributions of the paper include: a conceptualized task model of qualitative coding, a publicly available qualitative coding dataset, and a detailed evaluation of qualitative coding suggestion models via automated experiments and human subject studies.

**Questions For The Authors:**

see weaknesses.

**Reasons To Accept:**

- The conceptual task model of qualitative coding is well-motivated and each of the properties identified — data saturation, rare codes, annotation style —, are grounded on existing work.


- The Quoding dataset is a valuable contribution which will drive future research on the task of qualitative coding suggestion (QCS).


- The authors provided sufficient details of the different modeling approaches for QCS and rationalized their choices of models for each approach.


- The experiment setup is detailed and seems reproducible.


- The authors conducted detailed experiments presenting and analyzing results of automated experiments and human subject studies. The high-level insights from these studies reinforced the initial task model outlined by the authors.

**Reasons To Reject:**

- The paper fails to situate itself with respect to existing work. The related work section should be updated to differentiate the proposed approach with related work.

- The objective of the user study was not clarified. The authors mention evaluating the “effectiveness of the proposed QCS strategy.” However, what specific research questions they plan to address through the study is unclear. The author should clarify whether the evaluation was meant to be a thorough study of user experience with the proposed QCS method or simply meant to be a verification step.

**Reproducibility:**

4: Could mostly reproduce the results, but there may be some variation because of sample variance or minor variations in their interpretation of the protocol or method.

**Reviewer Confidence:**

4: Quite sure. I tried to check the important points carefully. It's unlikely, though conceivable, that I missed something that should affect my ratings.

**Typos Grammar Style And Presentation Improvements:**

- line 661-662: hard to read
- Section 5.4 mentions that the zero shot prompt can be found in the appendix. Could not find the prompt in Section A.4.4.


Presentation/Organization suggestion:
- Table 1 is not informative and can be moved to Appendix.
- Scatterplots from A.7 should be presented in the main paper with sufficient detail. In the current form, the analysis section is not self-contained.
- To make space, the authors can make the discussions in Section 5.3 and 6.1 more concise. The modeling approaches are mentioned multiple times: at the beginning of Section 5, section 5.3, and section 6.1 and could be reorganized to make the discussion crisp.
- Section 7.1 and 7.2 are very difficult to read and the key insights are not apparent. The insights should be highlighted.

---

> ### Author Rebuttal · Authors · 2023-08-27
>
> We would like to thank the reviewer for their comments and very detailed feedback. We appreciate the reviewer pointing out that the conceptual task model of qualitative coding we propose is well motivated and grounded in existing literature. We would also like to thank the reviewer for highlighting the detailed and reproducible nature of our experimental setup and for pointing out that the analysis of our experiments support our task definition and the properties of qualitative coding we study (i.e., data saturation, rare codes and annotation styles). Finally, we would like to thank the reviewer for their strong view of this paper’s excitement and soundness.
>
> We first respond to the reviewer’s main points and then address their presentation/organizational suggestions:
>
> *Main points*
>
> Firstly, as it stands, there has been little work on studying automated assistance in inductive qualitative coding. The Chandra et al. (2019) reference the reviewer cites studies the process of inductive coding in general and not an automated assistive technique in particular. Some of the most notable work in automatic inductive coding includes Guetterman et al. (2018) who developed a WordNet-based clustering technique to group semantically similar text spans to give researchers an idea of commonalities in their dataset before they started the coding process. We would be glad to clarify the Related Works section and add additional references to differentiate our work from previous studies which bring automated assistance to inductive coding.
>
> Secondly, we would like to clarify some confusion regarding the human evaluation. The main purpose of the human evaluation was to determine whether the best QCS models were suggesting relevant codes even when the suggestions didn’t agree with the original annotations from the Quoding dataset. This extra validation step was necessary because we noticed that, in many cases, the qualitative coders did not consider all possible code assignments for a text span. Our human evaluation demonstrated that, even though our systems didn’t always rank the original codes highest, they consistently ranked relevant codes at the top. Thus, our human evaluation was not a user study meant to investigate the implications of QCS methods on coding workflow and findings. We would be happy to clarify the purpose of the human evaluation in Section 7.2 and add a more detailed description in the Appendix.
>
> Finally, in this paper, we do not measure the impact of the QCS task and its different modeling paradigms in a practical setting (See previous paragraph). We believe that such an endeavor merits additional follow-up work to be explored in a human-computer interaction venue. The reason for a rigorous user study being beyond the scope of this paper is that we expect findings to be rich and diverse. For instance, in Rietz and Maedche (2021), the authors dedicate the entire publication to study the implications of retrieving text spans related to a given code (the “dual” of QCS). We would be happy to expand this line of thought in the Conclusion as future work.
>
> *Presentation/organization suggestions*
>
> We thank the reviewer for their thoughtful and detailed suggestions on the paper’s presentation and organization. We would be happy to reorganize the Results section, as suggested by the reviewer, to make the key insights apparent. The key insights include: 1. The information-retrieval and zero-shot paradigms performing better than the commonly-used classification approach and 2. The statistically significant jumps in performance when model outputs are evaluated by humans. We are happy to apply the reviewer’s organizational suggestions to move Table 1 to the Appendix, to remove some redundant information in sections 5.3 and 6.1 and to place the scatterplots in the main paper to support the Analysis section. We are also glad to address the readability issue raised on lines 661-662 and to add the GPT3.5 prompt which we inadvertently deleted and which we include below
>
> """You are a helpful assistant that suggests qualitative codes for a qualitative researcher. The coders you can suggest are: [AVAILABLE_CODES], and None of the above. Which of the previous codes would you assign to the following excerpt from an interview with a woman at risk of cardiovascular disease (CVD): “Question: [QUESTION] Answer: [ANSWER]” """
>
> Thanks again to the reviewer for their constructive feedback and useful comments.

---

### Official Review · Reviewer_ZTy2 · 2023-08-06

**Soundness:** 4

**Excitement:**

4: Strong: This paper deepens the understanding of some phenomenon or lowers the barriers to an existing research direction.

**Paper Topic And Main Contributions:**

This study investigates the problem of qualitative coding: assigning descriptive labels to the text spans of a document. The authors propose a more assistive approach following an automated code suggestion taking into account a ranked list of previously suggested codes. At the same time, the authors define the problem as QCS: qualitative code suggestion and developed a dataset Quoding consisting of interviews from cardiovascular domains. The findings include identifying the effect of personalized annotation style, how taking into account facts like passage sequence, rare-code appearance makes an impact. The evaluation includes both automated scoring and human judgment.

**Questions For The Authors:**

n/a

**Reasons To Accept:**

-comprehensive study
-new contribution and development of resource
-verification of the proposed method assisted by human evaluation


**Reasons To Reject:**

-the workflow was difficult to follow in some cases. It would be useful to provide a diagram describing the overall workflow of the method.

**Reproducibility:**

5: Could easily reproduce the results.

**Reviewer Confidence:**

3: Pretty sure, but there's a chance I missed something. Although I have a good feel for this area in general, I did not carefully check the paper's details, e.g., the math, experimental design, or novelty.

---

> ### Author Rebuttal · Authors · 2023-08-27
>
> We would like to thank the reviewer for their comments and their feedback. We appreciate the fact that they highlighted the comprehensive nature of our work which proposes to take a more assistive approach to qualitative coding and which also includes a human evaluation. We would also like to thank the reviewer for pointing out that our work develops and releases the first qualitative coding dataset* (*Pending publication). Finally, we would like to thank the reviewer for their strong view of this paper’s excitement and soundness.
>
> We address the reviewer’s main point below:
>
> We would be happy to add a figure describing the overall workflow of our approach in the camera-ready version of the paper. This figure will present, at a high level, the collection of the interviews and the annotations carried out by the qualitative coders, the preprocessing of the annotations to create the Quoding dataset, the three methods we study to tackle the Qualitative Code Suggestion (QCS) task, and finally the human evaluation. We highly appreciate the suggestion and believe that these changes would improve the presentation and general readability of the paper.
>
> Thank you again for your feedback and your suggestions.

---

### Meta-Review · Area_Chair_rQDr · 2023-09-18

**Recommendation:** 4

**Metareview:**

The paper proposes a user-centric approach to assist qualitative researchers by providing automated agent generated code suggestions.

Reviewers agreed that:
1. The paper presented a comprehensive study
2. The conceptual task of qualitative coding is well-defined
3. The dataset is useful.

Originally reviewers WMQc and ULRd had some concerns about the connection between this and prior work, and the validity of the user study, which the authors sufficiently addressed in rebuttal.

I recommend accepting the paper into the Main track, and encourage the authors to add a figure illustrating the workflow, extending the related work session, and providing clarification to the human eval.

---

### Decision · Program_Chairs · 2023-10-07

**Decision:**

Accept-Findings

**Comment:**

The paper proposes a user-centric approach to assist qualitative researchers by providing automated agent generated code suggestions.

Reviewers agreed that:
1. The paper presented a comprehensive study
2. The conceptual task of qualitative coding is well-defined
3. The dataset is useful.

Originally reviewers WMQc and ULRd had some concerns about the connection between this and prior work, and the validity of the user study, which the authors sufficiently addressed in rebuttal.

I recommend accepting the paper into the Main track, and encourage the authors to add a figure illustrating the workflow, extending the related work session, and providing clarification to the human eval.